# Self-Regulation of Attention in Children in a Virtual Classroom Environment: A Feasibility Study

**DOI:** 10.3390/bioengineering10121352

**Published:** 2023-11-24

**Authors:** Carole Guedj, Rémi Tyrand, Emmanuel Badier, Lou Planchamp, Madison Stringer, Myriam Ophelia Zimmermann, Victor Férat, Russia Ha-Vinh Leuchter, Frédéric Grouiller

**Affiliations:** 1Swiss Center for Affective Sciences, University of Geneva, 1202 Geneva, Switzerland; remi.tyrand@unige.ch (R.T.); emmanuel.badier@unige.ch (E.B.); frederic.grouiller@unige.ch (F.G.); 2Laboratory of Behavioral Neurology and Imaging of Cognition, Department of Basic Neurosciences, University of Geneva, 1202 Geneva, Switzerland; lou.planchamp@etu.unige.ch (L.P.); mad.stringer.me@gmail.com (M.S.); myriam.zimmermann@unige.ch (M.O.Z.); 3Functional Brain Mapping Laboratory, Department of Basic Neurosciences, University of Geneva, 1202 Geneva, Switzerland; victor.ferat@unige.ch; 4Division of Development and Growth, Department of Paediatrics, Gynaecology and Obstetrics, Geneva University Hospitals, 1205 Geneva, Switzerland; russia.ha-vinhleuchter@hcuge.ch; 5CIBM MRI Cognitive and Affective Neuroimaging Section, Center for Biomedical Imaging, University of Geneva, 1202 Geneva, Switzerland

**Keywords:** attention deficit hyperactive disorder, EEG-neurofeedback, virtual reality, attention, fMRI, theta/beta ratio

## Abstract

Attention is a crucial cognitive function that enables us to selectively focus on relevant information from the surrounding world to achieve our goals. Impairments in sustained attention pose challenges, particularly in children with attention deficit hyperactivity disorder, a neurodevelopmental disorder characterized by impulsive and inattentive behavior. While psychostimulant medications are the most effective ADHD treatment, they often yield unwanted side effects, making it crucial to explore non-pharmacological treatments. We propose a groundbreaking protocol that combines electroencephalography-based neurofeedback with virtual reality (VR) as an innovative approach to address attention deficits. By integrating a virtual classroom environment, we aim to enhance the transferability of attentional control skills while simultaneously increasing motivation and interest among children. The present study demonstrates the feasibility of this approach through an initial assessment involving a small group of healthy children, showcasing its potential for future evaluation in ADHD children. Preliminary results indicate high engagement and positive feedback. Pre- and post-protocol assessments via EEG and fMRI recordings suggest changes in attentional function. Further validation is required, but this protocol is a significant advancement in neurofeedback therapy for ADHD. The integration of EEG-NFB and VR presents a novel avenue for enhancing attentional control and addressing behavioral challenges in children with ADHD.

## 1. Introduction

Paying attention is not always easy, and lapses can occur even with our best efforts. Some of these failures are minor, like forgetting someone’s name or missing an object in our visual field. But others can have disastrous consequences, like distracted driving that leads to car accidents [1]. While attention lapses are common, some people may have an amplified pattern of these failures, such as those with attention deficit hyperactivity disorder (ADHD). This disorder affects 5 to 7% of school-aged children [2] and can lead to academic failure, anxiety, depression, behavioral disorders, relational issues, and drug addiction [3]. Research has revealed that people with ADHD have reduced functional connectivity between frontal and parietal regions [4,5], and their brain activity shows increased levels of slow theta (4–8 Hz) activity and reduced alpha (8–12 Hz) and beta (13–30 Hz) activity [6].

Conventional treatments for ADHD rely on stimulant medications such as methylphenidate (Ritalin^®^) [7]. These drugs work by increasing levels of norepinephrine and dopamine in the brain, which are considered the neurobiological cause of ADHD. However, 44% of children do not achieve remission [8], many suffer from undesirable side effects [9], and most have difficulty maintaining benefits beyond two years of treatment [10]. In addition, adherence to therapy, which is a crucial factor in the success of any treatment, is often a difficulty for children with ADHD entering adolescence [11].

Alternative therapies to medication such as psychotherapy, diet, or mindfulness have been proposed to reduce the systematic use of stimulants in children. But suboptimal adherence or persistence in treatment regimens is a widespread concern among children grappling with attention-related challenges [12]. Various factors contribute to this issue in real-world scenarios, including individual factors like age, influence of parents or family members, social aspects in the context of group therapies, and overall motivation and interest in the prescribed treatment. Therefore, there is a pressing need for innovative therapeutic strategies that can be tailored to individual factors, resources, and specific needs.

Among these emerging therapies, electroencephalography-based neurofeedback (EEG-NFB) stands out as a promising approach. It offers a non-invasive, non-pharmacological method that addresses both healing and prevention while mitigating side effects and potentially providing lasting clinical benefits. Through EEG-NFB, patients learn to regulate their brain electrical activity through real-time feedback extracted from their EEG signals. The ratio between theta and beta rhythms of the EEG has shown encouraging results in teaching children with ADHD to regulate their brain activity and reduce symptoms in the long term [13,14]. Preliminary studies show that effects are maintained over follow-up periods of 6 and 24 months, with a tendency towards a greater decrease in hyperactivity and impulsivity symptoms after 24 months [15]. Moreover, as the reliance on a subjective assessment approach can be prone to patient and expert bias, there is an effort to develop new ways of informing clinical diagnosis and treatment effectiveness using objective symptom biomarkers, EEG being one method of interest [16,17]. In July 2013, the FDA (Food and Drug Administration) approved the use of TBR as a biomarker for ADHD diagnosis [18]. In the context of EEG-NFB for ADHD, TBR has undergone thorough investigation and has demonstrated specificity [19].

One of the significant limitations of current EEG-NFB approaches is that they require repetitive training sessions before stable and positive effects are achieved [20]. In practice, this corresponds to 30–40 sessions involving long procedures such as the placement of numerous electrodes and the testing of their signal, followed by repetitive computer tasks that are difficult to maintain even for adults. The typical tasks of NFB protocols have nothing to do with how we sustain attention in real-life situations and with the challenges that children with ADHD face in their daily routines, especially in the classroom, which probably limits the transferability of what is learned during the intervention.

To overcome these limitations, we propose an innovative EEG-NFB protocol that leverages a virtual classroom environment. This environment has been specifically designed to enhance the transferability of attentional control skills beyond the training sessions and to challenge children through a distraction-modulated setting. Our research hypothesis sought to validate the feasibility of this novel EEG-NFB protocol in the context of virtual reality for school-aged children, with the overarching goal of increasing motivation and reducing the total number of required sessions.

In this report, we demonstrate the feasibility of this groundbreaking protocol on a small group of healthy children aged 6 to 11 years. This study presents a significant step forward in neurofeedback treatment by validating an innovative and engaging protocol that holds promising potential for future testing in children with ADHD. This research opens up new horizons for the field, highlighting the effectiveness of a playful and immersive approach to neurofeedback therapy. It is important to note that due to the small sample size, all statistical analyses should be interpreted with caution. The statistical tests are included primarily for informational and qualitative purposes rather than quantitative conclusions.

## 2. Methods

### 2.1. General Procedure

The experimental protocol included a total of 12 sessions: 8 sessions of EEG-NFB training based on theta/beta ratio (TBR) and performed in an immersive virtual classroom environment, 2 sessions of neuropsychological assessments (pre- and post-training), and 2 sessions of EEG-fMRI (pre- and post-training). The details of each session are summarized in Figure 1. Sessions were distributed in a maximum duration of 12 weeks with an optimal rate of one session lasting for one hour per week.

### 2.2. Participants and Dataset

Participants in this study were recruited through flyers posted in multiple academic buildings within the university. Prior to their participation, written consent was obtained from the parents or guardians of the children. The children themselves were informed about the study’s objectives and procedures and were assured that they could choose to discontinue their involvement at any time without facing any adverse consequences.

The study included a total of six healthy children (i.e., typically developing, without ADHD diagnosis), with an average age of 9.46 years (SD: 1.23) and including one female participant. However, two children were excluded from the final analysis due to technical issues during the neurofeedback training sessions. Additionally, another child was unable to complete the pre- and post- fMRI sessions due to artifacts caused by a dental appliance, but was still included in all other parts of the study.

In sum, a total of four children took part in EEG-NFB training sessions performed in a VR cave (sessions 3 to 8), and three of them also underwent EEG-fMRI scans before and after these sessions (sessions 2 and 11, respectively).

The core of the protocol comprises the sessions conducted within the VR cave. These sessions encompass EEG-NFB training, as well as additional exercises like a sustained attention task (go/nogo task) and a calculation task (see Section 2.4.1 and Section 2.4.2). For clarity and conciseness, we will provide a detailed description of these specific sessions below. In these sessions, this preliminary study analyzed the children’s performance during the sustained attention task, as well as changes in the TBR during the NFB training task (Section 2.4.1). For the pre- and post-imaging sessions, we only analyzed the resting-state runs to evaluate data quality and the transfer run to investigate the effect of attentional training on EEG power and brain activation. This study was conducted in accordance with the Declaration of Helsinki and approved by the Research Ethics Committee of the Geneva University Hospital (CCER 2020-02642).

### 2.3. Materials

#### 2.3.1. Virtual Classroom

General Aspects

The virtual classroom environment was designed to closely mimic the daily reality of schoolchildren. The child was seated at a real school desk and immersed in the virtual environment, which included a classroom-like setting, such as a whiteboard, clock, and alphabet book on the wall (as depicted in Figure 2), as well as other students, a teacher, and a multitude of audio and visual distractions.

Detailed equipment

The virtual classroom has been set up in the Brain Behavioral Laboratory Immersive-System (BBL-IS). This virtual reality (VR) cave system features four screens presenting seamless and perspective-coherent 3D images. The projection screens are made of four semi-transparent acrylic coated screens (left, front, right, ground). Their dimensions are 2.4 m × 2.4 m, except for the front screen, which is wider (2.8 m). These large screens enable the VR environment to be video-projected all around the user, enhancing the immersive experience. To do so, four Barco F70-4K8 video projectors (Barco, Kortrijk, Belgium) were used with a resolution of 2560 × 1600 pixels and a refresh rate of 120 Hz. Each video projector is located behind its corresponding screen to prevent the user from obstructing the projections. The VR cave also includes a motion tracking system composed of ten Vicon Bonita infrared cameras. Vicon Tracker 3.0 software is used to track motion of the reflective markers positioned on the head of a child. Vicon Tracker has a built-in Virtual Reality Peripheral Network server (https://vrpn.github.io/, accessed on 25 October 2023), which streams head positions to the VR simulation at a frequency of 240 Hz. To deliver 3D spatialized sound, we used a 5.1 sound system (Denon AVR-1312, and KEF KHT2005.2 speakers). The setup includes two computers. The first computer, featuring dual Intel Xeon E5645 CPUs, an Nvidia Quadro K5200 GPU, and 18 GB DDR3 RAM, serves for managing all recording systems such as motion tracking, EEG recording, and real-time processing. The second computer, featuring dual Intel Xeon Gold 6144 CPUs, dual Nvidia Quadro RTX 6000 GPUs, and 128 GB DDR4 RAM, is dedicated to running the VR simulation. An MMBT-S Trigger Interface Box (Neurospec AG, Stansstaderstrasse, Switzerland) is connected to this latter computer to synchronize the VR simulation with the EEG system. The code used was written in C-sharp and is available (https://github.com/ebadier/NeurospecTriggerBox-Unity, accessed on 25 October 2023). Finally, a 4-button Response Box (Current Designs Inc., Philadelphia, PA, USA) was used to allow children to answer questions during tasks performed in VR with a time resolution of 0.8 ms.

VR stimulation

The VR simulation has been developed using the Unity game engine. We used the assets described in Table 1 from the Unity Asset Store.

Script animation

The virtual scene was designed to closely mimic real-life classroom conditions and challenge the children’s attention levels in a progressive manner by incorporating distraction periods. During these periods, the distractions were randomly introduced during specific time blocks of the VR simulation (every 30 s). Each task began with a 30 s distraction-free block. The frequency of distractions was proportional to the child’s level of concentration, as determined by the normalized TBR. The less focused the child was, the fewer distractions occurred, with a minimum of one distraction during distraction periods and a maximum of one distraction every 3 s if the child was fully focused. This dynamically adjusted the distraction rate to provide a highly personalized, engaging, and ecologically valid training experience. It also ensured sustained participant engagement, averting boredom during high focus and frustration during attention lapses. When not in distraction periods, the virtual characters displayed slow-motion animations, but remained quiet. The distractions were randomly chosen from those described in Table 2 and only one distraction could occur at a time during distraction periods.

To add to the illusion of something happening in the classroom, all characters in the scene would look in the direction of the distraction when it occurred. When the distraction ended, they would return to their slow-motion animations in a seamless manner.

#### 2.3.2. EEG Setup for EEG-NFB Sessions

The EEG was acquired from 32 sintered Ag/AgCl multi-electrodes mounted in an elastic cap adapted to children’s head sizes according to the 10–20 montage (EasyCap GmbH, Wörthsee, Germany). One of the electrodes was dedicated to ECG recording and was positioned on the participant’s back on the left under the shoulder plate. Electrodes were equipped with 10 kOhm resistors (20 kOhm for ECG) to avoid induced currents during simultaneous fMRI recordings [21]. To ensure a good electrical contact between the scalp and the electrodes, high-chloride abrasive gel was used after degreasing the scalp with isopropyl alcohol. Before starting the recording, the scalp impedances were lower than 30 kOhm. The EEG was acquired at 500 Hz using a 32-channel BrainAmp MR Plus amplifier (Brain Products GmbH, Gilching, Germany, RRID:SCR_009443) connected to the cap with a bundled ribbon cable.

The real-time processing of the EEG was performed using OpenVibe (Acquisition Server & Designer 2.2.0, used to acquire EEG and process data, respectively, RRID:SCR_014156, https://scicrunch.org/resources, accessed on 25 October 2023), an open-source software for brain–computer interfaces developed by INRIA (http://openvibe.inria.fr, accessed on 25 October 2023). The following computations were executed on a time-moving Hanning window of 3 s, which was updated every second. First, an average reference was computed by averaging the signal of every EEG electrode and subtracting this average from each electrode. Then, a fast Fourier transform was performed on electrode Fz and its spectral power was computed in frequency bands corresponding to theta (4 to 7.5 Hz) and beta oscillations (13 to 19 Hz). Finally, the TBR was computed and the results were sent to the Unity application with the use of a TCP/IP connection. Muscular artefacts due to jaw movements that may corrupt the beta oscillation power were detected by computing the beta oscillation power over the temporal electrode T7. If the beta power of electrode T7 exceeded a threshold of 3 times the beta power of electrode Fz, then the TBR sent to the Unity classroom application was set to zero and the subject was prompted to relax their facial muscles.

#### 2.3.3. Simultaneous fMRI-EEG Setup

Simultaneous EEG-fMRI acquisitions were performed in a 3T MRI (Siemens Magnetom Trio or Siemens Magnetom Prisma Fit, Philadelphia, PA, USA). For the EEG, the same setup as for the EEG recordings in a virtual reality environment outside MRI was used but the sampling rate was set to 5 kHZ and a synchronization box was used to assure a 10 MHz alignment to the MRI scanner clock. These changes in EEG recording were necessary for the correction of the artefacts caused by the MRI on the EEG signal. The EEG cap was connected with a short bundle ribbon cable to the amplifier, located at the back of the MRI bore [22]. The EEG amplifier and the wires were immobilized using sandbags and memory foam pillows to avoid any vibrations. The child’s head was also restrained with memory foam cushions to limit movement.

A T2*-weighted single-shot gradient-echo echo-planar image sequence was used for functional MRI acquisition (TR = 1760 ms, TE = 30 ms, flip angle = 90°, 32 interleaved axial slices of 3 mm with an inter-slice gap of 0.8 mm, 64 × 64 acquisition matrix, in-plane resolution = 3 × 3 mm^2^, bandwidth = 2004 Hz/Px, GRAPPA acceleration factor of 2). A 3D T1-weighted structural image with a 1 mm isotropic resolution was also acquired.

### 2.4. Behavioral Tasks

#### 2.4.1. Neurofeedback Training Task

Calibration Phase

Because the raw TBR of each child is unique, it needs to be normalized (ranging between 0 and 1) in order to adjust for distractions and neurofeedback tasks based on each child’s attentional capabilities. A calibration phase was designed to identify the optimal minimum and maximum values for normalization. This phase consisted of four steps: two relaxing and two focusing periods interleaved, each lasting 20 s. For the relaxing periods, the child was prompted by the experimenter to relax, while they were prompted to make a mental calculation for the focusing periods. During these steps, raw TBR values were collected and sorted, and any values generated by muscle artefacts were removed (see Section 2.3.2). The minimum value was calculated as the minimum value of the sorted array, and the maximum value was calculated as the average value of the last quarter of the sorted array.

Helicopter display

During each EEG-NFB training, children practiced two runs, with each run lasting for 3 min. Those runs were interleaved with cognitive tasks. EEG data were transmitted using the TCP/IP protocol to OpenVibe (RRID:SCR_014156) to estimate in real-time the TBR (see Section 2.3.2). During these runs, a helicopter was displayed on the whiteboard of the classroom. The height of the helicopter flight was dependent on the TBR recorded in real-time using the following formula:*h*(*t*) = *h*(*t* − 1) + *vd*(*t*) + *gd*(*t*);
where *h* is the height of the helicopter, *vd* is the vertical displacement calculated as [*nfb* × *speed* × *dt*], *gd* is the gravity displacement computed as [−0.5 × *speed* × *dt*] and was applied to the helicopter every frame to simulate gravity, *dt* is the time elapsed, *nfb* is a per-subject normalized TBR value within the range [0;1], and *speed* is a constant of 10 m/s.

To maintain the helicopter within the boundaries of the task, the height (*h*) was clamped during each frame to the minimum and maximum height values, corresponding to the bottom and top of the whiteboard.

The child was asked to find strategies to regulate their attention by trying to take off in the helicopter to the highest elevation possible and maintain flying at the highest elevation possible.

Calculation task as baseline

In this task, children were engaged in solving simple arithmetic calculations. To accommodate the wide range of competencies among children aged 6 to 11 years, we dynamically monitored the performance of the child. The accuracy was computed continuously during the task. When a child achieved a success rate of less than 70%, the difficulty level of the calculations remained the same. Only addition and subtraction operations involving a single digit were presented (e.g., 9 + 4 or 7 − 3). On the other hand, if a subject surpassed a 70% success rate, the task became more challenging. Apart from addition and subtraction with single digits, the calculations now included addition and subtraction with two digits, as well as multiplication with one digit (e.g., 34 + 8 or 5 × 9). During each trial, the calculation was presented to the child, who was then required to select the correct answer from three given possibilities. This interactive format actively engages participants and encourages them to focus their attention on the arithmetic. Therefore, we utilized the initial 20 s of this cognitively engaging task to establish the TBR baseline (see Section 2.6.3).

#### 2.4.2. Sustained Attention Task

The sustained attention task is a commonly used task for the diagnosis of attention disorders [23]. Children were asked to press a button whenever a letter appeared, except for the letter “X”. In each task block, the rate of occurrence of “X” was 25%. A block comprised 20 trials, and each trial lasted 2 s (randomized values around: 750 ms for fixation cross, 250 ms for letter stimuli, and 1000 ms for response time). In total, there were six blocks of each condition performed in interplay. If the child did not respond to a correct letter or took more than one second after the presentation of the stimuli to respond, it was considered as an omission error, and if the child pressed the button after the letter ‘X’, it was considered as a commission error.

#### 2.4.3. Neurofeedback Transfer Task

The NFB transfer task serves as the final assessment in the protocol, evaluating the children’s ability to self-regulate their brain activity without the presence of feedback. During this task, children were instructed to apply the attention regulation strategies learned during EEG-NFB training, but without receiving any visual feedback. Periods of regulation were interleaved with periods of rest every 30 s, comprising a total of 6 blocks per condition.

#### 2.4.4. Behavioral Data Analysis

Satisfaction survey

We assessed the children’s satisfaction via a binary response questionnaire given at the end of the protocol. The results are reported in Table 3.

Sustained attention task

To evaluate the children’s motivation throughout the sessions, their performance in the sustained attention task was measured in terms of accuracy and reaction times. Reaction time data were cleaned by excluding error trials (mean 10.8% ± 1.7%) and trials with reaction times faster or slower than three times the standard deviation from the mean RTs of each individual subject (mean 2.9% ± 0.8%).

### 2.5. EEG and fMRI Data Preprocessing

Three children underwent EEG-fMRI scans before and after EEG-NFB training sessions in the virtual reality cave (sessions 2 and 11, respectively). In this preliminary study, only resting-state runs were used to assess data quality, and the transfer run was used to investigate the impact of training on brain activation.

#### 2.5.1. EEG Data in MRI

Gradient artefacts were corrected using a hybrid mean and median average subtraction method, which prevents head movement artefacts from entering the template [24]. For each time point in the template, the values of the *L* neighbouring artifacts were sorted and the *K* minimal and maximal values excluded from the averaging process (*L* = 30, *K* = 6). This method is less sensitive to motion and is well-adapted to children’s datasets, as it eliminates large outliers from the moving averaged gradient artefact template. Then, the EEG was low-pass filtered with a finite impulse response (FIR) filter with a cut-off frequency of 70 Hz and downsampled to 500 Hz. After detecting the QRS complex in the ECG channel, pulse artefacts were removed using a hybrid mean and median moving average subtraction method (*L* = 40, *K* = 5).

Additional preprocessing steps were carried out with MNE-python (RRID:SCR_005972, [25]) and were similar to those described in Section 2.5.2.

#### 2.5.2. EEG Data in VR

EEG recordings were preprocessed offline using MNE-python (RRID:SCR_005972, [25]). First, the recordings were bandpass filtered, restricting the frequency range to 1–40 Hz using an FIR filter. A notch filter was also applied to eliminate 50 Hz line noise. Second, to ensure data quality, segments containing artifacts and channels with poor signal were manually annotated and excluded from subsequent analyses. To remove noncerebral artifacts such as eye blinks and muscle activity, independent component analysis (ICA) decomposition was performed using the Picard algorithm [26]. Finally, bad channels were interpolated using spherical spline interpolation. Recordings were re-referenced to the average reference for further analysis.

#### 2.5.3. fMRI Data

Both structural and functional MRI data were preprocessed using fMRIPrep 21.0.2 [27] on Nipype 1.6.1 [28].

Anatomical data

T1-weighted (T1w) structural images were corrected for intensity non-uniformity (N4BiasFieldCorrection, ANTs 2.3.3, [29,30])*,* and they were skull-stripped (antsBrainExtraction.sh, ANTs 2.3.3) using OASIS30ANTs as the target template. Brain tissue was segmented into cerebrospinal fluid (CSF), white matter (WM), and gray matter (GM) (fast, FSL 6.0.5., [31]). Volume-based spatial normalization to standard spaces (MNIPediatricAsym:cohort-3, an unbiased template for pediatric data from the 4.5 to 18.5 y age range) was performed through nonlinear registration (antsRegistration, ANTs 2.3.3) using brain-extracted versions of both the T1w reference and the T1w template.

Functional data

For each subject, the functional MRI (i.e., the two resting-state runs and the NFB transfer run) underwent the following pre-processing steps. First, a reference volume and its skull-stripped version were generated using a custom methodology of fMRIPrep. Rigid-body head motion parameters with respect to the BOLD EPI reference were estimated (mcflirt, FSL 6.0.5., [32]) before any spatiotemporal filtering. BOLD runs were slice-time corrected (3dTshift, AFNIR, RID:SCR_005927, [33]). The BOLD time series were resampled onto their original, native space by applying the transformations to correct for head motion. The BOLD reference was then co-registered to the T1w (mri_coreg, FreeSurfer; flirt, FSL) with the boundary-based registration cost function. Co-registration was conducted with six degrees of freedom.

The data underwent additional preprocessing using the AFNI software v.10.7 (RID:SCR_005927, [33]). A spatial smoothing was applied using an 8 mm FWHM Gaussian kernel to reduce surrounding noise [34]. Finally, a linear regression was performed to eliminate nuisance variables such as estimated motion parameters, the first-order temporal derivatives of motion parameters, squared motion parameters, and mean time courses of the cerebrospinal fluid and white matter signals. These nuisance variables were further adjusted for linear and higher-order polynomial trends [35,36].

### 2.6. EEG and fMRI Data Analysis

#### 2.6.1. Resting-State fMRI Networks

To identify brain networks, we applied a probabilistic independent component analysis (ICA) approach using the FMRIB Software Library’s MELODIC toolbox v.6.0 on the full dataset, obtained by temporally concatenating across subjects, sessions, and runs [37] to create a single four-dimensional dataset decomposed into 20 independent components. We categorized the resulting ICs into three groups (akin to [38,39] based on visual inspection: (1) ‘real’ (bilateral networks consistent with known anatomical and functional circuits), (2) ‘duplicate/unclear’ (bilateral or noisy map hat overlapped with real networks), and (3) ‘noise-related’ ICs (random speckle patterns and rim patterns). To produce final maps, we applied a posterior probability threshold of *p* < 0.5 (equal loss placed on false positives and false negatives) by fitting a Gaussian/Gamma mixture model to the histogram of intensity values [40]. Similarly, to evaluate data quality at the individual level, we also performed ICA for each subject separately, concatenating pre- and post- resting-state fMRI runs.

#### 2.6.2. EEG Microstates

The EEG microstate analysis was conducted using Pycrostates [41]. For each recording, the local maxima of the global field power (GFP), which represent segments of the EEG data with the highest signal-to-noise ratio [42], were identified. These GFPs were then subjected to modified k-means clustering (polarity-independent) with 100 repetitions, selecting the clustering result with the highest global explained variance. A predetermined value of k = 5 microstate maps (cluster centroids) was estimated for each recording, considering the exploratory nature of the study and drawing from previous literature [43,44].

The estimated maps at the recording level were subsequently merged and subjected to the same clustering algorithm to extract k = 5 topographies that best represent the dataset. These k = 5 global dominant topographies were then fitted back to the original EEG data. In this procedure, time points were assigned cluster labels (i.e., microstate topographies) based on spatial correlation analysis. Each time point was assigned to the topography with which it had the highest absolute spatial correlation. To ensure temporal continuity, a smoothing window of 50 ms was applied, adjusting the central time point’s correlation with a smoothing factor of 10.

#### 2.6.3. TBR in EEG-NFB Sessions

Frequency bands were obtained by applying spectral decomposition techniques using the fast Fourier transform algorithm [45]. Specifically, we focused on the theta and beta band to further compute TBR. The average TBR on the Fz electrode was calculated across all eight sessions for each subject for the two helicopter task runs and the first 20 s of the calculation task, which served as the baseline.

Furthermore, we calculated the evolution of the TBR relative to the baseline for each session and every subject as follows:$${TBR}_{relative\;to\;baseline}={mean\;TBR}_{helicopter}-{mean\;TBR}_{baseline}$$

Finally, the TBR evolution was averaged across all subjects, enabling us to gain insights into the collective trends and variations in TBR across the entire child cohort.

#### 2.6.4. Neurofeedback Transfer Task

EEG

The data were divided into 10 s epochs and categorized as either ‘Rest’ or ‘Regulation’, depending on the corresponding block period of the NFB transfer task. Epochs containing residual artifacts were manually identified and excluded from further analysis. Similar to the TBR computation during EEG-NFB sessions, we used the fast Fourier transform algorithm to extract the theta and beta frequency bands. Subsequently, we calculated the average TBR on the Fz electrode separately for the ‘Rest’ and ‘Regulation’ epochs.

fMRI

The linear regression model described above (see Section 2.5.3) included an additional regressor representing the onset of ‘Rest’ and ‘Regulation’ blocks. Contrasts comparing the ‘Regulation’ to ‘Rest’ conditions were computed for each child and subsequently entered into a second-level analysis using a one-sample analysis of variance (ANOVA). A statistical threshold of *p* < 0.05, uncorrected for multiple comparisons, was applied in the analysis.

## 3. Results

### 3.1. Acceptation, Satisfaction, and Sustaining Motivation

The children’s acceptance of the EEG and fMRI sessions as part of the protocol was complete. Out of the six children who started the EEG-NFB training sessions, all successfully completed the protocol, resulting in a 100% completion rate. However, two participants had to be excluded from the results analysis due to technical issues (as outlined in Section 2.2). Additionally, one more family attended the initial neuropsychological assessment session but later opted not to participate, citing time constraints as the reason for their withdrawal.

At the end of the experiment, a satisfaction survey was administered to the six participating children. The results, displayed in Table 3, indicate that all children reported enjoyment of the study. Furthermore, five out of the six children expressed satisfaction with the virtual reality experience and noted improvement in their concentration abilities. However, it was found that 83% of the children were unable to consistently apply the concentration strategies learned during the sessions in their daily classroom setting.

To assess the maintenance of the children’s motivation during the EEG-NFB sessions in the VR, we plotted their performance in the sustained attention task throughout the sessions in terms of accuracy and reaction times. The integration of distractions was a critical aspect of the experimental design, as it mimicked the conditions of a real-life classroom and challenged the participants’ attentional capacities. As such, we also studied the effects of these distractions on the children’s performance. The results of this study showed that the four participating children had an average accuracy of 89% (±5%) on the sustained attention task across the eight runs (Figure 3A). Their response times seemed to be influenced by the presence of distractors, showing an average response time of 504 ms (±64%) without distraction compared to 526 ms (±67%) when distractions were present.

Taken together, these results suggested the stability of the children’s performance across the multiple sessions, indicating a sustained level of motivation to complete the experimental protocol. Additionally, the findings highlight the impact of the distraction period, confirming the expected distraction effect.

### 3.2. Feasibility in Children: Quality Assessment of Simultaneous EEG-fMRI Data

To ensure the feasibility of our protocol, we examined the quality of the EEG and fMRI data acquired during simultaneous recordings (sessions 2 and 11). High-quality data are essential for future studies to evaluate the effect of neurofeedback on brain activation and connectivity. To assess data quality, we focused on resting-state runs to identify established maps in both fMRI (i.e., resting-state networks) and EEG data (i.e., microstates). These maps are recognized for their sensitivity to noise and motion artifacts, rendering them valuable tools for the detection of data quality concerns.

First, we applied the ICA approach to fMRI data to identify the well-known and documented resting-state networks [46]. At the group level, we observed seven resting-state networks that exhibited the typical bilateral pattern of ‘real’ resting-state networks (Figure 4A). We categorized the remaining networks as either ‘duplicate/unclear’ or ‘noise-related’ ICs (Appendix A). At the individual level, the ICs were less clear, likely due to the smaller amount of data and participant movements during the scanning session (680 time points versus 2040 for the grouped data). However, we were still able to identify the dorsal fronto-parietal network involved in attention processes [47] at both the group and individual levels (Figure 4B).

In addition, we evaluated the quality of the EEG data acquired simultaneously with fMRI by performing microstate analysis at the group level. Similar to ICA, microstate analysis is a technique that enables the examination of spatiotemporal patterns within EEG recordings through k-means clustering. This method involves decomposing the multichannel EEG signal into a series of quasi-stable states, where each state is characterized by a specific spatial distribution of scalp potentials known as a microstate map. This analysis provides valuable insights into the temporal dynamics and spatial patterns of the EEG data and can be used in a future study to evaluate the impact of EEG-NFB training on these dynamics. As depicted in Figure 4C, decomposition of the signal resulted in five maps. Four of these maps were consistent with the microstates traditionally described in the literature [43], while the fifth topography corresponded to residual cardioballistic artifacts originating from the simultaneous EEG-fMRI setup [48].

### 3.3. Potential of EEG-NFB Combined with VR: Preliminary Findings

Finally, we assessed the potential of this innovative protocol. First, we examined the EEG data recorded during the EEG-NFB training sessions. The average TBR on the Fz electrode was computed per session for each child, relative to their baseline. The progression of this value across the eight training sessions for the four children is depicted in Figure 5 (left panel). A notable tendency emerges when comparing the first training period (sessions 1 to 4) to the second (sessions 4 to 8). Three out of four children demonstrated a decrease trend in their TBR during the later stage of training, indicating a learning process for controlling brain activity (see Figure 5, right panel).

Second, we measured the children’s ability to regulate their attention during the NFB transfer task, i.e., inside the MRI scanner and without any feedback. Our objectives were to evaluate their proficiency in controlling brain activity and to investigate the corresponding neural correlates. To do so, the children were instructed to alternate between periods of rest and periods of concentration. In the EEG data, we observed a consistent tendency to decrease in TBR at the Fz electrodes for all three children (Figure 6A). Interestingly, we observed that the children did not exhibit a uniform pattern of TBR modulation. Child 1 displayed a decrease in theta activity, while child 2 displayed an increase in beta activity, and child 4 showcased the ability to modulate both theta and beta rhythms.

Simultaneously, in the fMRI data, we found a significant reduction in brain activation within regions associated with the default-mode network and sensorimotor cortices, which occurred during the attention regulation periods compared to the resting periods (although not corrected for multiple comparisons) (see Figure 6B and Appendix A). This finding indicates a cognitive engagement in the children and potentially suggests an improved ability to filter out external distractions.

## 4. Discussion

The answer to our initial question is yes, our innovative EEG-NFB protocol, combined with a virtual reality environment, is feasible in children and shows potential suitability for children with ADHD. We propose that it is likely to be accepted and attractive to this vulnerable population. Additionally, to the best of our knowledge, this is the first study to combine these technologies to propose a non-pharmacological alternative for treating attention deficits. This report also provides comprehensive information about the VR environment developed by our team, specifically designed to introduce EEG-NFB training to school-aged children between 6 and 11 years old.

Among the children who began the EEG-NFB training, the final participation rate in the study was 100%. These findings demonstrate that once enrolled in the study, and particularly once the EEG-NFB training commenced, participants and their families exhibited a strong interest and willingness to fully engage in the intervention. While comparisons with other studies may be challenging due to the small cohort size utilized in this study, the compliance results appear to surpass those reported in other NFB studies involving children [49].

The satisfaction results of this study are promising, but it is important to consider the limitations within its context. Firstly, our satisfaction scale was limited to multiple choice responses, and our work could have benefited from the use of a numerical scale for a more precise quantification of children’s satisfaction. Secondly, the satisfaction survey indicated that the children encountered difficulties in applying the concentration strategies they had learned for controlling the helicopter in a real classroom environment. In future research, we intend to enhance the neurofeedback intervention by incorporating a pre-session tutorial. This tutorial will provide children with a deeper understanding of attention processes and guide them on effectively utilizing these strategies to achieve their goals. By addressing these limitations, we aim to further optimize the intervention’s efficacy and improve overall satisfaction among the participants.

The quality of both the MRI and EEG recordings was deemed adequate for conducting the intended group analyses and evaluating the effects of our innovative EEG-NFB protocol. However, at the individual level, the analysis was somewhat constrained by residual movement artifacts. Nonetheless, in future studies, enhancing data quality, particularly during resting runs (i.e., when no active engagement of the participant is required), could be achieved by using visual stimulation with abstract shapes. It has been shown that this approach can improve children’s compliance during prolonged sessions, such as resting-state fMRI, thereby improving data quality [50].

To conclude on the therapeutic potential of our protocol, it is essential to acknowledge the limitations of our study, namely the small sample size of our group of children and the absence of attentional or neuropsychological disorders among the four participants. Indeed, there are inherent complexities and individual differences in children with ADHD that may impact the feasibility and effectiveness of our EEG-NFB protocol. Despite these limitations, it is encouraging to observe that both EEG and fMRI data consistently align in the same direction, indicating an improvement in TBR. The reproducibility of this improvement even in the absence of feedback highlights the children’s capacity to acquire the skill of controlling their brain activity. Furthermore, this is also confirmed by neuroimaging data showing a decrease in activation in regions of the default-mode network, which operates antagonistically to the task-positive network, also referred to as the attention network [51,52]. However, it is essential to recognize that this study serves as a preliminary investigation using state-of-the-art brain imaging technology; to comprehensively evaluate the potential therapeutic impact, future research will need to be conducted, specifically in children diagnosed with ADHD.

An important aspect of this study involves the individual variations in TBR modulation observed during the transfer run. This variability underscores the well-known complexity inherent in the treatment of children with ADHD. The syndrome often manifests as multidimensional, with significant heterogeneity in symptoms and electrophysiological profiles [53,54]. These findings reinforce the notion that personalized interventions, potentially based on individualized electrophysiological markers, are crucial for effective therapy of this pathology.

## 5. Conclusions

To the best of our knowledge, this study represents a pioneering effort in combining EEG-NFB and VR to provide a relatively short and playful therapeutic protocol aimed at teaching children with ADHD to regulate their own brain activity.

Once enrolled, the full rate of engagement in the intervention program also shows that the children largely accepted and enjoyed the program. While the efficacy of the protocol has yet to be validated, the preliminary findings are exceptionally promising.

To summarize, this novel EEG-NFB protocol combined with VR holds significant potential as a valuable tool for enhancing executive functions and behavioral skills among children with ADHD. Considering the power of brain plasticity [55], this innovative intervention could have a significant impact on the well-being and development of people with ADHD. In perspective, this study not only highlights the initial success of the protocol but also emphasizes the need to explore playful and immersive interventions in the treatment of childhood pathologies [56].

## Figures and Tables

**Figure 1 bioengineering-10-01352-f001:**
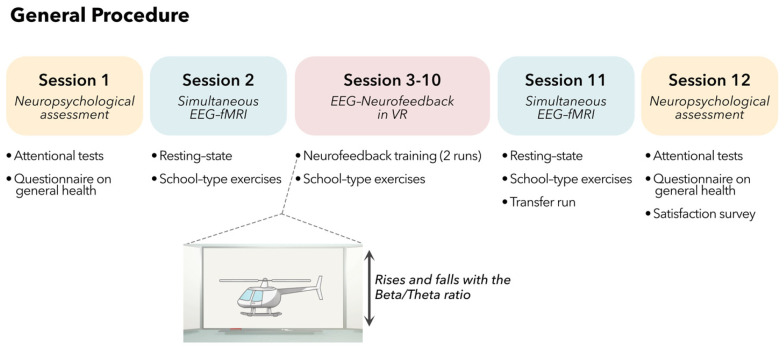
Protocol overview.

**Figure 2 bioengineering-10-01352-f002:**
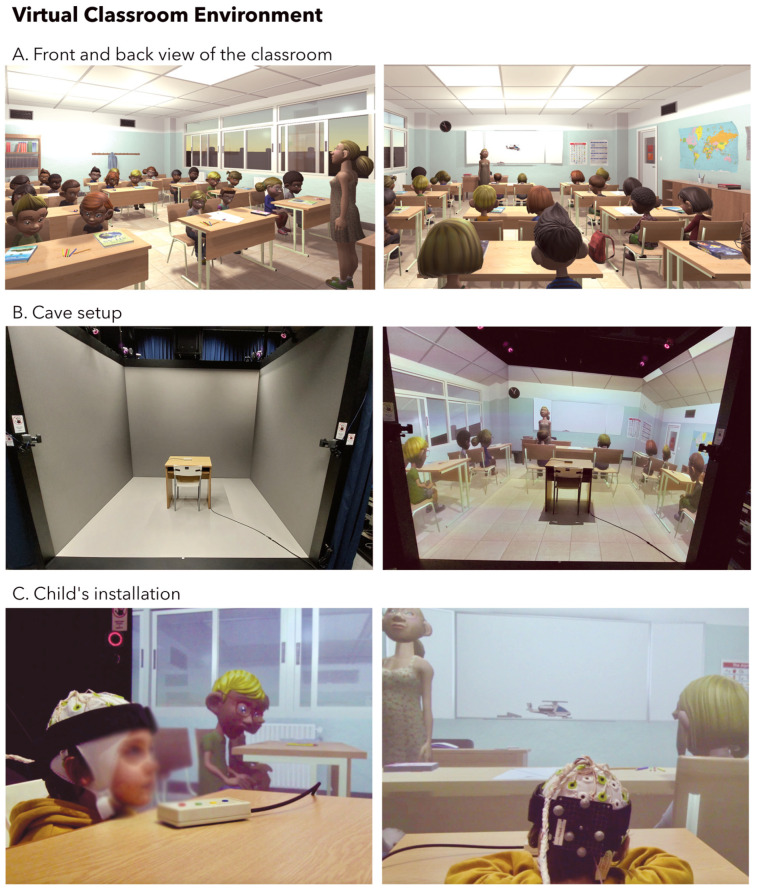
Virtual classroom environment. (**A**) Front and back view of the virtual classroom. (**B**) Cave setup with screens off (**left**) or on (**right**). Projectors and motion sensors are positioned around the three vertical walls. (**C**) Child’s installation within the virtual reality cave. An EEG cap is placed on the child’s head, complemented by a headband equipped with infrared sensors. These sensors enable the visual scene to dynamically synchronize with the child’s movements, enhancing the overall realism of the virtual reality experience.

**Figure 3 bioengineering-10-01352-f003:**
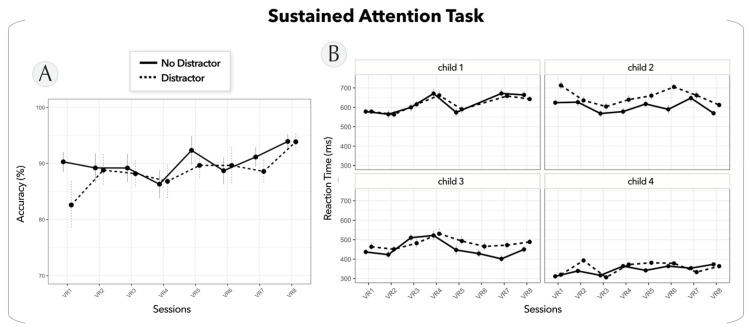
Sustained attention task. (**A**) Task accuracy average across the four children and (**B**) reaction time for each child during sessions in the VR classroom environment for the conditions without distraction (solid lines) and with distractions (dotted lines). Error bars reflect the standard errors of the mean.

**Figure 4 bioengineering-10-01352-f004:**
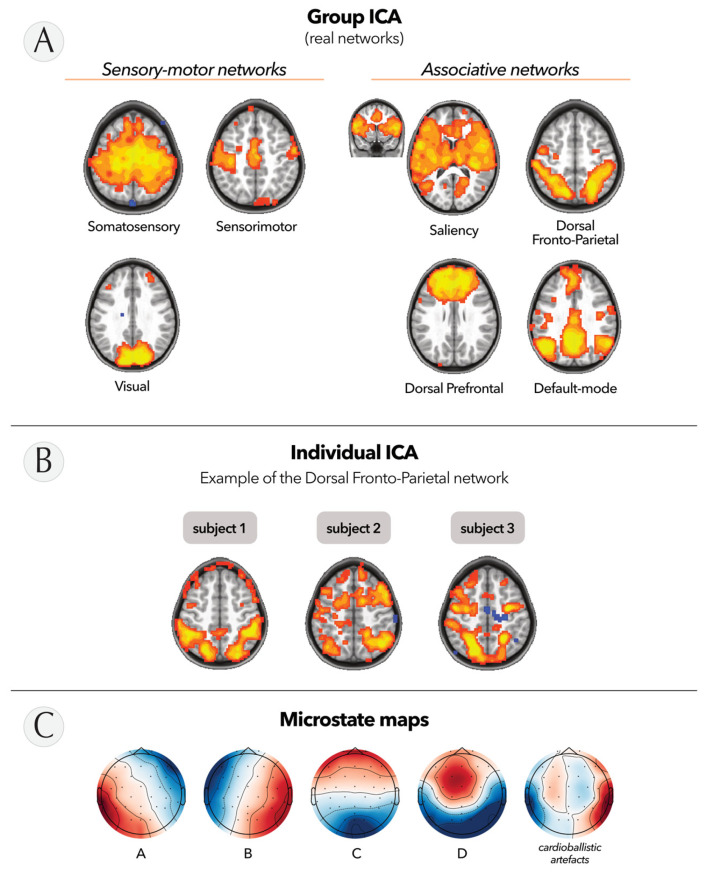
fMRI and EEG data quality. (**A**) Resting-state networks identified in fMRI with ICA at the group level and overlaid on axial or coronal slices. Warm colors correspond to positive correlations and cool colors to negative correlations. (**B**) Associated dorsal fronto-parietal network from the subject level. (**C**) Microstate maps identified in EEG acquired simultaneously with fMRI at the group level.

**Figure 5 bioengineering-10-01352-f005:**
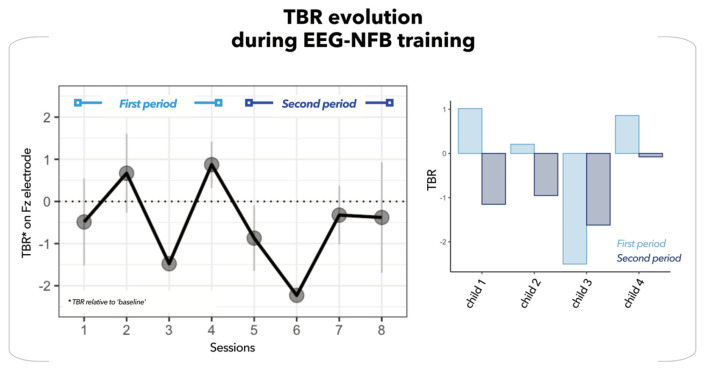
TBR evolution during EEG-NFB training. (**Left panel**) Average TBR (black dot line) on Fz electrode for the eight EEG-NFB training sessions. This TBR value corresponds to the TBR during helicopter task after subtracting the baseline. (**Right panel**) TBR values extracted for each child from the first (light blue bar) and second (dark blue bar) periods of the protocol.

**Figure 6 bioengineering-10-01352-f006:**
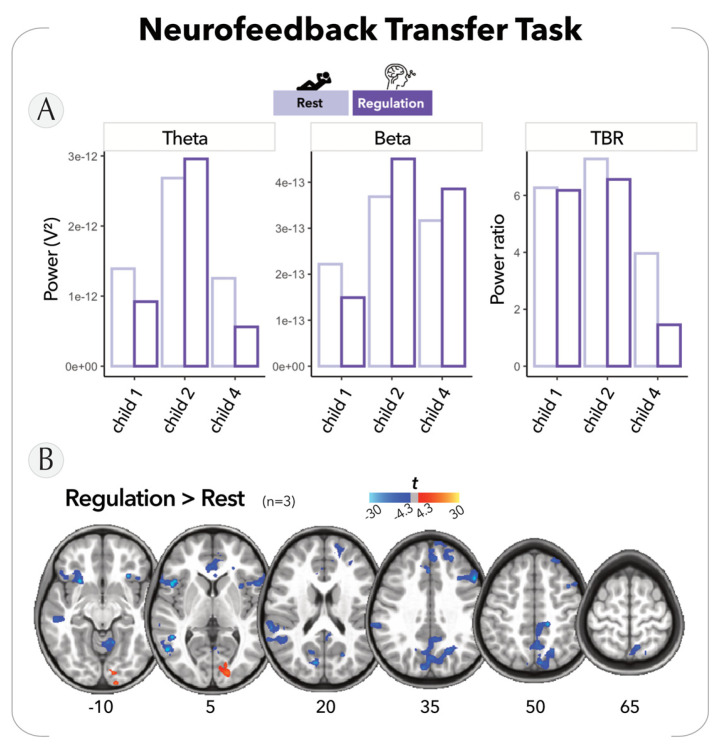
Preliminary results on neurofeedback transfer task. (**A**) Theta and beta power and TBR on Fz electrode extracted for each child from the ‘Rest’ (light purple) and ‘Regulation’ (dark purple) periods. (**B**) Statistical maps of the contrast of ‘Regulation vs. Rest’ overlaid on axial slices (*p* < 0.05, uncorrected). The color-bar shows T-value task-related activations (yellow/red) and inactivations (blue).

**Table 1 bioengineering-10-01352-t001:** Unity assets used in the VR simulation.

Name	Description	Asset Link
School classroom	Classroom used as the 3D environment of the VR simulation	https://assetstore.unity.com/packages/3d/characters/humanoids/2-toon-people-116917, accessed on 25 October 2023
Toon characters	20 ‘toon kids’ (10 boys and 10 girls) sat at their desks in pairs, all around the (participant) child. The latter was placed in the center of the virtual classroom sitting at a real school desk (which was the center of the CAVE system)	https://assetstore.unity.com/packages/3d/characters/humanoids/humans/toon-kids-55945, accessed on 25 October 2023
Toon people	2 ‘toon people’ were used to represent the virtual mistress and school’s headmaster	https://assetstore.unity.com/packages/3d/characters/humanoids/2-toon-people-116917, accessed on 25 October 2023
Everyday motion pack—free	Package of animations (idle, sit, walk, talk) used to animate virtual character bodies	https://assetstore.unity.com/packages/3d/animations/everyday-motion-pack-free-115067, accessed on 25 October 2023
SALSA lip sync	Plugin used to animate virtual character faces, to generate various random head and gaze directions, eye blinks, and “look at target” behaviors. It also allowed virtual characters’ lips to move in sync with virtual character’ speech (audio-to-speech)	https://assetstore.unity.com/packages/tools/animation/salsa-lipsync-suite-148442, accessed on 25 October 2023

**Table 2 bioengineering-10-01352-t002:** Panel of distractions.

Character	Type of Stimulation	Description
Mistress	Audiovisual	– walk – cough – yawn – speak randomly among 11 sentences (voices recorded from one adult female)
Headmaster	Audiovisual	– enter and exit the classroom – walk – cough – yawn – speak randomly among 6 sentences (voices recorded from one adult male)
Kids	Visual	– raise their hands
	Audiovisual	– speak among 53 sentences (voices recorded from one girl and one boy)
Other	Audiovisual	– phone on the mistress’s desk is vibrating
	Audio	– end of class bell rings
	Audiovisual	– noisy insects (fly, dragonfly, butterfly) fly in front of the whiteboard

**Table 3 bioengineering-10-01352-t003:** Satisfaction survey.

Questions	Yes	No	Sometimes
Did you like the VR classroom sessions?	83%	0%	17%
Were you happy to come and do the experiment?	100%	0%	0%
Do you feel that you have improved your ability to fly the helicopter?	66%	0%	33%
Did you have a strategy for flying the helicopter? *	100%	0%	0%
Do you use these strategies to focus in class?	17%	83%	0%
Do you feel like you are able to focus better?	83%	0%	17%

* Examples of strategies used: “I recited the multiplication tables”; “I focused on the image of the helicopter”; “I focused on a detail of the classroom”. The shaded cells correspond to the majority response over the 6 children who completed the protocol. Note that 2 of them were removed from the rest of the analyses due to technical issues (see Section 2.2).

## Data Availability

The method section has provided a detailed and reproducible description of the virtual classroom setup. Given the innovative nature of this study, data access can be granted upon reasonable request. The scripts and data will be accessible once the results are confirmed with a larger cohort.

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
