# Peer review of "Self-Regulation of Attention in Children in a Virtual Classroom Environment: A Feasibility Study"

_bioengineering, 2023, doi:10.3390/bioengineering10121352_

Round 1

Reviewer 1 Report (Previous Reviewer 1)

Comments and Suggestions for Authors

I appreciate the authors' response to my comments and their effort in revising the paper. I also continue to be impressed with the technical work and think this sort of approach, if done carefully, could positively impact the community. Unfortunately, the paper, in my opinion, continues to draw conclusions far beyond the data gathered, and the authors did little to respond to that comment. Given the potential and likely interest in follow up work, I would be willing to reconsider, subject to the journal's policy on multiple revisions, but I recommend rejection.

Specific comments on what I see as major issues that should be resolved before publication.

1. Participants - I thank the authors for clarifying that all the participants were healthy, or maybe 'typically developing'. As I mentioned in the previous review, this decreases the potential impact of the study.

Also, the use of 'completion rate' is possibly misleading since it ignores both the 7th 'family' that opted out early (456-458) and the data loss in 2 (3? still not clear) of the participants. The authors may feel confident that their collection issues are solved, but I don't see what evidence they have provided. 

Later in their response about MR, the authors also mention two other studies they have been involved in both of which had a 50% dropout rate on just MR (on a quick read so please correct me). These papers are not cited in the article. Why not use this experience, and others from other groups, as context for the dropout rates? 50% is much closer to what I would have expected. 

It is a little hard to believe that 6 (or even just 3 or 4, I'm still unsure) children donned an EEG cap multiple times over many days, sat in VR for extended periods, and went into an MR twice without any complaining, and the authors should provide strong evidence for such a claim.

As a participant feasibility study, the presentation is somewhat misleading and does not address the ADHD population. However, the authors make the unsupported claim in the first sentence of their discussion that "The answer to our initial question is yes, our innovative EEG-NFB protocol, combined with a virtual reality environment, is feasible in children with ADHD and is likely to be accepted and attractive to this vulnerable population." This is very misleading and makes claims well beyond the modest results the authors presented that their task pipeline worked in 3 (or 4?) out of 6 (or 7?) cases.

2. Task - I again thank the authors for some clarification on the task, and it has some good potential, but is missing some validation. As long as it's clearly stated, the decision not to include children with ADHD is potentially a good one, if the task had been clearly shown to measure valid outcomes. Roughly, the task has three parts - behavior, questionnaires, and physiology. None of these are satisfactory.

Questionnaires - The authors have acknowledged that their 'questionnaire lacks some essential elements'. They do not, however, provide specifics about plans to address that lack in the main paper.

Behavior - CPT's are notoriously easy to implement poorly. I have every confidence that the authors' implementation is fine, but they provide no evidence that I can find. A comparison with a 'standard CPT', like the NIH toolbox, is missing. The limited data they show have some wide ranges in reaction time (300ms versus 600ms), only report 'accuracy' instead of the various omission and commission rates, and report misleading chi^2 tests. The helicopter control with TBR description is appreciated, and it is clear how it is moving. The Brownian-type movement is a little unusual and would benefit from some reporting about how the parameters were tuned.

Physiology - The authors chose not to report validation of their EEG-training paradigm. Given the authors admitted knowledge of products that "...superficially resemble EEG and neurofeedback..." validating their design should be of the utmost importance to the study. What evidence do the authors have that the height of the helicopter in the game is related to theta/beta ratio in the brain?

The authors use of microstates, or not, and MR, or not, are of secondary importance to the question of if the training they ask the children to do is actually targeting what they want to train.

3. Statistics - I disagree with all the authors statistical conclusions. There should be no statistical tests on outcomes in a feasibility study, and they are misleading, at best. The chi^2 tests in particular are not '..informative rather than providing definitive proof..' as the authors state, but just misleading. For example, treating session as a factor in the main effect disregards correlation between sessions per participant that is clear in figure 3. 

The naive approach to statistics comes up again later with a note that '3 out of 4' did something (line 528)- if I flipped a coin and got 3 out of 4 heads, so what? These statistical issues are glaring and lead the authors to later draw conclusions that are unacceptable even stated as 'just trends'.

I'm always willing to reassess, but without removing the misleading statistics and talk about 'trends', the authors leave the paper open to mis-interpretation and the potential for using their data out of context. For example, their abstract claims - "Pre- and post-protocol assessments via EEG and fMRI recordings suggest improved attentional function." This is not supported by their data, and can easily be taken out of context. 

Author Response

Reviewer 2 Report (Previous Reviewer 2)

Comments and Suggestions for Authors

Dear authors,

Many thanks for responding to the previous comments and enhancing this pioneering paper. The additional text adds to the quality of the paper and also adds clarity. For example, the additional images of the cave setup enable a clear understanding of how it looks with the screens on and off.

Given the small number of subjects for the trial, the increased use of tentative terms such as tendancy, provides a more appropriate wording and consideration of the findings.

The English used is of a high quality and the only recommendation is to adjust the wording on line 195 - "This dynamically adjusted the distraction rate to provide a highly personalized..."

Finally, I do hope that you continue to further investigate this very interesting line of research - I think you are on to something special.

Author Response

We thank the reviewer for his/her positive feedback on the additional content. We are pleased to hear that these enhancements contribute to the overall quality and clarity of the paper. We agree that employing more cautious language, given the small number of subjects in the study, is more appropriate.

We addressed the suggested adjustment on line 195 to improve the wording.

The reviewer’s encouragement to further explore this line of research is inspiring, and we share his/her enthusiasm for the potential it holds.

Reviewer 3 Report (Previous Reviewer 3)

Comments and Suggestions for Authors

Authors did a good job in revision.

Author Response

We thank the reviewer for his/her positive feedback.

Round 2

Reviewer 1 Report (Previous Reviewer 1)

Comments and Suggestions for Authors

Thanks to the authors for further clarifying the scope of the study. The paper seems OK as a methods description, and I look forward to seeing the larger study.

This manuscript is a resubmission of an earlier submission. The following is a list of the peer review reports and author responses from that submission.

Round 1

Reviewer 1 Report

Comments and Suggestions for Authors

The authors present a feasibility study, presumably in preparation for a larger scale study. The overall goal is to address building attention skills in children with ADHD, which drives broad interest. The wide range of technologies and tests in young children also make it interesting to consider feasibility by itself. While the authors note that this is a feasibility study, they unfortunately attempt to draw conclusions and, e.g.,  fit statistical models. I recommend refocusing the presentation on the feasibility aspect and avoiding unwarranted conclusions.

Major changes recommended to stay within the scope of feasibility.

1) Participants: 6 (or maybe 7? it is unclear) participants were recruited, and 3 completed the study set. I would expect completion rate to be a major outcome of a feasibility study, but the authors seem to gloss over '..technical issues during neurofeedback training...' (line 96). What were these technical issues and have they been addressed? Also there is no description of disease state for the participants. The authors should please clarify if the children had ADHD diagnoses. If not, then they should also justify how these data will generalize to the ADHD group. Or, if so, please report how ADHD diagnosis was made.

2) Methods: The methods would benefit from some clarification. Given that a use of this paper will be to cite it in future work, methods lapses rise to a major issue. I list the questions I still have, but the authors should please place more focus on the methods overall.

2a) Methods CAVE: The CAVE hardware description is appropriate, but a picture of the experimental system would improve figure 2 and how timing is handled is a little unclear, especially since reaction (response) time is an outcome - what are the latencies of the various systems? The description of the paradigm is also too sparse, and could not be replicated from the information provided. Please also discuss why the distraction rate is linked to the TBR and the advantage over, e.g., playing the same VR for all of the participants. A supplemental video would be helpful, if the journal allows.

2b) Methods EEG: The hardware description is clear. Some discussion is missing of how the participants accepted wearing EEG on 8 sessions, especially given the abrasive gel. The authors use (at least) two EEG processing pipelines: online for real-time feedback and offline for later analyses. Was the TBR estimate similar from both pipelines? Similarly, an estimate of TBR from the EEG-MRI data would help make the connection between the measures. Also, beyond a brief mention, analyses like microstates and grand averages are out of scope for a feasibility study.

2c) Methods MR: The MR hardware description is also clear, and the analysis pipeline is clear. Please add a note on how well the participants accepted the MRI procedure that can be daunting for adults, yet alone young children. E.g., did a guardian accompany the participant in the MR room?

2d) Methods Task: Please specify task details. What CPT did the authors use? If it was developed in house, then please describe it in more detail (e.g. fonts and exact timing) and provide some validation for it, like a comparison with, e.g., the NIH toolbox. How exactly did the helicopter move with TBR? Linear proportion would make it skip all over, so there was likely some sort of physics model - if so what was it? The questionnaire is lacking, as the authors note later, not only missing a Likert scale, but also some questions. I would also have expected a parent/guardian questionnaire to accompany it, e.g., the Vanderbilt NIHCQ. There are many video game questionnaires the authors might adapt, e.g., there are many 'immersion' questionnaires that could be used or adapted. Also, a note on the planned statistical approach is ok, but unless the authors are creating a power analysis, it is not very interesting.

3) Results: The results occasionally stray from focus on only the feasibility of the design. I will make a sub-list again concentrating on the results section.

3a) Survey results: The table is appreciated, but distressingly lacking questions like 'How often do you play video games?' or 'How much do you like helicopters?'. I see this as a major fault with the study, but I understand the complexity of mixing task and questionnaire design. The authors should please describe how they plan to control for comparisons between children who love helicopters and play 'Helicopter Commander' for hours every day, with those who think helicopters are dumb and just want to play football?

3b) Performance results: While I understand the authors want to see early results, none of the statistical presentation is appropriate with only 3 or 4 participants. The individual breakdown in figure 2B is more convincing that the task design is acceptable to the participants than the group average in 2A. Statistical tests across 4 participants are not interesting (line 418 and 419). Stability is a reasonable thing to look for, but the authors should please avoid any groupwise tests, at least until they perform the larger study. Lines 511-518 appear in the 'discussion' but are actually results, and are an OK description. But, why claim 100% participation rate? 7 participants were recruited (including the one mentioned on line 512), and 3 completed the full paradigm. 3/7 is not 100%, and 40% completion rate is a number the authors should be aware of.

3c) MR results: The fMRI results are most convincing at the individual level (e.g. Figure 4B or 6A) and show good promise for future measures. The authors should please avoid discussions of groupwise comparisons, and instead attempt to answer a question like 'Does the MRI pick up data in regions of interest?' (almost surely, yes) or 'Will children with ADHD accept two MR sessions?' The latter is most important. Similarly, the microstate map (4c) is too advanced for feasibility, and I recommend removing the microstate analysis, except possibly as a note for future measures in the discussion.

3d) EEG-NFB Results: To belabor the point - all the references to groupwise comparisons should be removed. Phrases like 'Three out of four children demonstrated a decrease...' (line 473) are statistically nonsensical and indicate nothing. The authors should please use the space to validate their measure, e.g. compare TBR measured from the several EEG pipelines. I also would like clarity on why 'normalized TBR' is not in the interval [0,1]. What do the authors mean by 'normalization'? Is there evidence that the training variable (helicopter height) is related to the normalized TBR?

4) In the discussion, despite the authors' claims, there are many EEG training paradigms, from home based models you can buy off the shelf to MRI studies that are readily located. The specific paradigm may be original and the authors should be commended for pulling together the potential for gathering these data on children with ADHD. But, the discussion should place it in context. For example, some of the publicly available products that make claims about ADHD have had very little impact and, arguably, given EEG-attention training a negative reputation. How do the authors think this work will fit into such a landscape? No conclusions about groupwise differences are appropriate in the discussion, and even trends are not meaningful. Again, the authors should please concentrate on feasibility - can they perform the planned data collection? Also, this is clearly intended as an early paper for a larger future study. Could the authors please provide details on the planned study?

Overall, reviewing this review, I realize that it is quite critical, and I hope that the authors take the criticism as constructive as it was intended. A careful study along the lines of what is proposed would be of great interest, a technical marvel, and potentially benefit many people.

Comments on the Quality of English Language

The grammar is nearly all fine, although there are some minor typos that a quick run through a word processor would find more reliably than me.

Reviewer 2 Report

Comments and Suggestions for Authors

Thank you for this very interesting paper on your research. And, as a parent with a child who has ADHD, I read it with additional attention! Here are a few observations:

39. Perhaps a citation to support the value of attention and car accidents.

54. Perhaps a sentence or two mentioning the variety of alternative interventions other than pharmacological. This would then provide a foundation and springboard  into NFB and for your innovative study using VR.

77. Your use of 'healthy children' doesn't appear to be explained - was there are reason for this? Perhaps it is to just develop the protocol and as a baseline.

Have you considered the dimension of cognitive workload to your studies? There is no need to incorporate a response to this question in your paper.

91. Fig. 1 helpfully clarifies the text. However, a more complete description of the actual activities in Sessions 3-10 would provide more insight. For example, was it just to make the helicopter fly? (230) Was any attempt made to capture the strategies which the subjects used to maintain concentration? This then raises the question about how the skills developed will assist learning transfer to the real world.

Section 2.3 - you provide a thorough description of the instruments and technology.

The research you conducted demonstrates a detailed level of attention to the protocol and during the procedures - you did well.

408. Some brief qualitative description of the strategies developed and then used or not used by the children in daily classroom settings would add further insight.

557/567 - Panksepp (2007) discusses the value of play in diminishing ADHD might you link to this?

Limitations - a double blind trial would enhance this study - perhaps this is something you are considering for future research?

You have demonstrated the potential of this innovative protocol and your pilot shows good promise for further investigations.

Reviewer 3 Report

Comments and Suggestions for Authors

It is an overall well written paper with appropriate structure and clarity. However, I do not see the descriptions of ethical consideration of research and recruiting procedure in the manuscript. Also I have the following comments for improvement of paper in terms of communication and delivery of the content better to readers.

1. the authors need to report on the recruiting process of participants. Who are they? How were they recruited? What is the relation between authors and participants? Did researchers obtain informed consent by guardians of children participants?

2. It would be clearer if authors provide research hypothesis.

3. Why did authors measure and analyze TBR? Providing basis on the selection of such indicator for attention seems necessary.

4. Providing a figure or image of experiment setting for simultaneous EEG-fMRI acquisitions would be helpful.

5. In line 434-436, “To ensure the feasibility of our protocol, we examined the quality of the EEG and fMRI data acquired during simultaneous recordings (sessions 2 and 11). High-quality data is essential for future studies to evaluate the effect of neurofeedback on brain activation and connectivity”

I think it would be helpful if authors provide evaluation criteria of the quality of their data in more detail.
